# Recent Advances in Chemoresistive Gas Sensors Using Two-Dimensional Materials

**DOI:** 10.3390/nano14171397

**Published:** 2024-08-27

**Authors:** Jae-Kwon Ko, In-Hyeok Park, Kootak Hong, Ki Chang Kwon

**Affiliations:** 1Division of Chemical and Material Metrology, Korea Research Institute of Standards and Science (KRISS), Daejeon 34113, Republic of Korea; 2Department of Analytical Science and Technology, Graduate School of Analytical Science and Technology (GRAST), Chungnam National University, Daejeon 34134, Republic of Korea; 3Department of Materials Science and Engineering, Chonnam National University (CNU), Gwangju 61186, Republic of Korea

**Keywords:** two-dimensional materials, gas sensors, nanostructure, chemoresistive gas sensors, heterostructure

## Abstract

Two-dimensional (2D) materials have emerged as a promising candidate in the chemoresistive gas sensor field to overcome the disadvantages of conventional metal-oxide semiconductors owing to their strong surface activities and high surface-to-volume ratio. This review summarizes the various approaches to enhance the 2D-material-based gas sensors and provides an overview of their progress. The distinctive attributes of semiconductor gas sensors employing 2D materials will be highlighted with their inherent advantages and associated challenges. The general operating principles of semiconductor gas sensors and the unique characteristics of 2D materials in gas-sensing mechanisms will be explored. The pros and cons of 2D materials in gas-sensing channels are discussed, and a route to overcome the current challenges will be delivered. Finally, the recent advancements to enhance the performance of 2D-material-based gas sensors including photo-activation, heteroatom doping, defect engineering, heterostructures, and nanostructures will be discussed. This review should offer a broad range of readers a new perspective toward the future development of 2D-material-based gas sensors.

## 1. Introduction

In recent years, gas sensors have been increasingly employed across various fields, including environmental monitoring, medical diagnostics, and industrial safety [1,2]. Given the complexity of indoor/outdoor gas environments and human exhalation, gas sensors must achieve high gas sensitivity, selectivity, and fast response/recovery. Various gas sensors have been developed including optical, electrochemical, pellistor, field-effect transistor (FET)-type, and chemoresistive gas sensors. Among these various types of gas sensors, the chemoresistive gas sensor offers several key advantages such as high gas sensitivity, rapid response/recovery, long operational lifetime, etc. Furthermore, their simple structure facilitates miniaturization, resulting in low costs and making them highly suitable for various applications. The performance of gas sensors can be evaluated based on the 4S criteria: sensitivity, selectivity, speed, and stability. Although extensive research has been conducted on fabricating various types of metal oxides in diverse morphologies to achieve high performance for target gases, there is still growing demand for further improvements of these performance metrics.

Semiconducting metal-oxides (MOSs) have been widely utilized as conventional gas-sensing materials due to their easy preparation and high stability in atmospheric conditions. However, these sensors face significant challenges that limit their performance and applicability. Notably, MOS sensors exhibit poor selectivity, being sensitive to a wide range of volatile organic compounds (VOCs), which compromises their ability to detect specific gases accurately. Their operation requires high temperatures (200–400 °C) to achieve high responsivity and rapid gas response/recovery, resulting in increased power consumption. Furthermore, MOS sensors are highly dependent on the degree of relative humidity, which can significantly impact their accuracy and stability. Additional limitations include long stabilization times, signal drift over time, slow recovery rates, and limited lifespans. These inherent drawbacks underscore the need for continued research and development to enhance the selectivity, energy efficiency, and overall performance of MOS-based gas sensors. Addressing these challenges is crucial for advancing the practical implementation of these sensors in various applications, from environmental monitoring to industrial safety and healthcare diagnostics [3,4,5]. Since the discovery of graphene, two-dimensional (2D) materials have garnered significant attention for their unique thickness-dependent physical and chemical properties. The extensive surface area and numerous dangling bonds of two-dimensional structures result in high surface activity, providing significant advantages for gas adsorption in sensor applications [6,7]. Recently, the family of 2D materials has expanded to include graphene, black phosphorus, transition metal chalcogenides (TMCs), and layered metal oxides, all of which have shown potential in gas sensing. Numerous studies have focused on 2D materials for gas sensing, employing various strategies to enhance the sensing properties of 2D-material-based gas sensors to meet increasing practical demands [8]. However, compared to conventional metal-oxide-based gas sensors, 2D-material-based gas sensor still have limited commercialization due to low performance in terms of recovery, long-term stability, detection limit, and selectivity. Thus, it is essential to explore recent developments in 2D-material-based gas sensors, including sensor types, sensing mechanisms, and influencing factors, to determine a way to further improvement in performance.

This review explores diverse 2D-material-based gas sensors, including graphene, transition metal dichalcogenides (TMDs), and their heterostructures, focusing on their distinctive physicochemical properties and sensing mechanisms. Recent advancements in synthesis techniques and sensing performance for volatile organic compounds (VOCs) are highlighted, demonstrating these materials’ high sensitivity and exceptional selectivity. The review provides critical insights into material fabrication methodologies, device integration strategies, and future research directions, contributing to the advancement of this rapidly evolving field in gas-sensing technology.

## 2. 2-Dimensional (2D) Materials for Chemoresistive Gas Sensors

Chemoresistive gas sensors operate based on charge transfer and conductivity modulation mechanisms, with their performance determined by changes in electrical resistance upon exposure to target gases. Key performance indicators include gas response, response/recovery time, selectivity, and detection limit. Enhancements can be achieved through nanostructuring, doping, catalyst integration, and heterojunction architectures. While 2D materials offer high sensitivity and rapid response due to their large surface areas, they face challenges such as low selectivity and slow desorption rates. Research focuses on improving these aspects through advanced nanostructures, surface functionalization, and photo-activation methods. Practical application requires testing in real-world environments and improving long-term stability to integrate these sensors into portable, wearable, and miniaturized systems. (See Figure 1).

### 2.1. Operating Mechanism

The performance of chemoresistive gas sensors is typically determined via the conductivity ratio before and after exposure to target gases, which is closely related to the major carrier concentration and mobility. In the case of metal-oxide-based semiconductors such as SnO_2_, ZnO, CuO, WO_3_, and Co_3_O_4_, they operate through the interaction of target gases with chemisorbed oxygen (O_2_) on the surface of sensing materials, a process influenced by temperature. In the air, O_2_ molecules capture electrons from the metal oxides, forming various O species depending on the temperature: O_2_^−^ (<100 °C), O^−^ (100 °C~300 °C), and O^2−^ (>300 °C). These interactions create an electron depletion layer in *n*-type semiconductors and a hole accumulation layer in *p*-type semiconductors, resulting in high resistance in the air. For the *n*-type gas sensors, such as those based on metal oxides like SnO_2_ and ZnO, resistance typically increases when exposed to oxidizing gases like NO_2_ [18]. This occurs because oxidizing gases capture electrons from the conduction band of the *n*-type semiconductor, decreasing the number of free electrons and thus increasing resistance. For example, when NO_2_ molecules adsorb onto the surface of an *n*-type sensor, they accept electrons from the sensor material, leading to higher resistance due to reduced electron density in the conduction band. Conversely, the resistance decreases when the *n*-type gas sensors are exposed to reducing gases such as H_2_ or NH_3_. Reducing gases donate electrons to the conduction band, increasing the number of free electrons, resulting in a resistance decrease. In contrast, *p*-type gas sensors, such as CuO and Co_3_O_4_, exhibit a decrease in resistance when exposed to oxidizing gases. These phenomena originate from the removal of holes from the valence band of the *p*-type semiconductor, increasing the number of free electrons, resulting in a resistance decrease. When *p*-type gas sensors are exposed to reducing gases, the resistance increases. Reducing gases donate electrons to the valence band, which recombines with holes, reducing the number of holes and increasing resistance.

The charge transfer mechanism, one of the fundamental sensing principles in 2D gas sensors, is predicated on the electronic interactions between the sensing material and target gas molecules. Upon adsorption of gas molecules onto the sensor surface, electron transfer occurs at the interface between the adsorbate and the sensing material, typically a semiconducting oxide. Oxidizing gases such as NO_2_ extract electrons from the sensing material, whereas reducing gases like NH_3_ inject electrons. This charge transfer process induces alterations in the electrical conductivity of the sensing material, serving as the primary basis for gas detection. The polarity and magnitude of the conductivity modulation are contingent upon both the semiconductor type (*n*-type or *p*-type) and the chemical nature of the target gas. For example, exposure to oxidizing gases results in increased resistance in *n*-type semiconductors but decreased resistance in *p*-type materials. The efficacy of this mechanism is influenced by several parameters, including the binding energy at the gas–sensor interface, the electronic band structure of both the sensing material and the target gas, and operational variables such as temperature and humidity. The schematic illustration of these working mechanisms of both metal-oxide semiconductors and 2D-materials-based gas sensors is summarized in Figure 2.

The performance optimization of gas sensors is intrinsically correlated with their fundamental operating principles, encompassing chemisorption, charge transfer, and conductivity modulation mechanisms. Critical performance metrics include gas response, response/recovery time, selectivity, and detection limit, which collectively determine the sensor’s sensitivity, accuracy, and applicability. Enhancing these parameters involves strategies that exploit the underlying sensing mechanisms: surface area maximization through nanostructuring augments gas–material interactions; doping and composite materials modulate carrier concentrations; temperature optimization influences oxygen species formation; catalyst integration facilitates specific gas reactions; and heterojunction architectures leverage complementary *n*-type and *p*-type semiconductor characteristics. Furthermore, crystal structure and defect engineering impact carrier mobility, while surface functionalization enhances selective interactions. These methodologies, when systematically implemented, can significantly improve the four critical performance indicators. For example, increased surface area can enhance gas response and lower detection thresholds, while targeted surface functionalization can improve selectivity. Optimized nanostructures and catalysts can minimize response and recovery times. Elucidating the distinct behaviors of *n*-type and *p*-type semiconductors in response to oxidizing and reducing gases provides a foundation for developing advanced, application-specific sensing technologies. Through the synergistic integration of these strategies, researchers can engineer gas sensors with superior performance across various operational parameters, extending the boundaries of sensitivity, selectivity, and real-time monitoring capabilities in diverse environmental conditions [21].

### 2.2. Advantages and Disadvantages of 2D Materials in Gas Sensing

2D materials hold significant promise for gas-sensing applications due to their large specific surface areas and numerous active sites compared to metal-oxide semiconductors. Recent advancements in chemoresistive and FET-based gas sensors based on 2D materials have focused on various aspects such as microstructure, synthesis methods, sensing properties, and corresponding sensing mechanisms [22]. 2D materials, such as graphene, TMCs [23], and black phosphorus (BP) [24], exhibit outstanding properties across various scientific fields, making them ideal candidates for chemical sensors [25].

Key parameters characterizing gas-sensing performance include response (or sensitivity), selectivity, response/recovery times, and stability. The successful fabrication of graphene has sparked widespread interest in 2D materials due to their unique structure and excellent properties. With their nanoscale thin-layer structure, 2D nanomaterials offer large specific surface areas and distinct electrical properties. Gas adsorption on the surface of these materials affects their conductivity, enabling them to serve as gas-sensing materials capable of adsorbing and capturing specific single species of gas molecules, thereby exhibiting excellent gas-sensing properties. Gas sensors based on 2D nanomaterials demonstrate several advantages, including high sensitivity, rapid response speed, low energy consumption, and the ability to operate at room temperature. Despite these advantages, the development of 2D chemoresistive gas sensors faces significant challenges. While two-dimensional materials possess high specific surface areas, this characteristic paradoxically leads to low selectivity due to their reactivity with various gas molecules. Furthermore, these materials often exhibit slow response and desorption rates, necessitating external stimuli for effective desorption, and, in some cases, desorption remains inefficient even with external stimulation. Additionally, the formation of chemical bonds due to surface dangling bonds impedes proper gas adsorption and desorption processes. These issues collectively contribute to low gas sensitivity in such sensors. To address these challenges, ongoing research efforts are focused on several strategies. These include the development of nanostructures, heterostructures with other metals, surface functionalization techniques, and the application of photo-activation methods, such as UV irradiation. These approaches aim to mitigate the aforementioned problems and enhance the overall performance of gas sensors based on two-dimensional materials.

This requirement limits the integration of these sensors into portable, wearable, and miniaturized multi-sensor systems [26]. Additionally, the sensing mechanisms of 2D materials are complex and not fully understood, involving charge transfer, gas molecule diffusion, adsorption/desorption, and carrier transport/recombination processes. Current characterizations are primarily qualitative, necessitating more comparative experiments and theoretical simulations. Moreover, practical application is hindered by the reliance on controlled environments rather than real-world complex gas backgrounds, such as human breath, which can affect accuracy and sensitivity due to interference and environmental factors [27]. To overcome these issues, testing on actual samples and enhancing long-term stability are essential. These efforts will be crucial in advancing the practical use of 2D-material-based gas sensors in diverse and more practical applications.

## 3. Recent Advances in Chemoresistive Gas Sensors with 2D Materials

Since graphene has been discovered and demonstrated by Geim and co-workers [6], 2D materials have gained tremendous attention because of their excellent physical, chemical, and mechanical properties. Among the excellent properties of 2D materials, the large specific surface area and large number of active sites for the physisorption and chemisorption of gas molecules facilitate the possibility of 2D materials as channel layers in resistive gas sensors. Here, we provide the recent advances in chemoresistive gas sensors based on 2D materials in this chapter: (i) graphene analog materials, (ii) TMCs, and (iii) a hybrid structure of 2D materials. Finally, we will also cover the heterostructure of the 2D materials to enhance the gas response and selectivity.

### 3.1. Graphene Analogues

#### 3.1.1. Graphene

Graphene, with its atomic carbon lattice, shows promise in optical, electrochemical, and mechanical applications, including its use in gas sensors due to its high surface-to-volume ratio and modifiable surface chemistry [25,28,29,30]. However, structural defects can impede its performance, necessitating defect engineering strategies such as atomic doping and functionalization to enhance gas interaction and sensor response [31,32]. Challenges persist in achieving rapid response and recovery properties. Kwon and Lee et al. have recently reported their findings on enhancing the gas response and selectivity of graphene gas sensors through *n*-type doping with ethylene amines via vapor-phase molecular doping [33]. Utilizing chemical vapor deposition (CVD)-grown graphene n-doped with ethylene amines, their sensor configuration, depicted in Figure 3a, was fabricated. Conventionally, CVD-grown graphene on SiO_2_/Si wafers tends to exhibit a p-doped state due to surface impurities and moisture/oxygen adsorption on the SiO_2_ layer [34]. To counteract these charge impurities, the SiO_2_/Si wafer surface underwent treatment with hexamethyldisilazane (HMDS). Employing amine functional groups—ethylene diamine (EDA), diethylene triamine (DETA), and triethylenetetramine (TETA)—for vapor-phase molecular doping provided a robust, uniform, and non-destructive doping method [35]. These amine groups, known for their electron-donating properties, facilitated charge transfer, inducing *n*-type doping in graphene. This n-doping rendered the graphene channel electron rich, thereby fostering electrostatic interactions with oxidizing gases such as NO_2_. Upon adsorption of NO_2_, hole carriers are generated, akin to p-doping. By doping the n-channel graphene sensor with three types of ethylene amines, the primary carrier changed from holes to electrons in pristine graphene, thus facilitating the response to oxidizing gases through electrostatic interactions (Figure 3b, left). The distinctive response of the n-doped graphene sensor to different target gases necessitates dynamic current measurements at zero-gate bias for NO_2_ detection (Figure 3b, right).

In atmospheric conditions characterized by complex gas mixtures, the ability to selectively detect target gases is crucial for minimizing spurious signals. Figure 3c,d illustrate the selectivity and summarized response of bare graphene, respectively. Selectivity testing was conducted with 50 ppm concentrations of NO_2_, SO_2_, and NH_3_ gases, revealing responses of 14%, 3%, and −0.3%, respectively, for pristine graphene. Notably, owing to its lower oxidizing strength compared to NO_2_, the response of SO_2_ gas appears diminished relative to that of NO_2_ gas. Conversely, DETA-doped graphene exhibited responses of −77%, −17%, and 0% to NO_2_, SO_2_, and NH_3_, respectively (Figure 3e,f), showing significantly higher responses to both NO_2_ and SO_2_ compared to pristine graphene. The response of graphene gas sensors to NO_2_ exposure, set at a concentration of 50 ppm for 5 min followed by a 30 min N_2_ purge, reveals distinct characteristics contingent upon molecular doping, as illustrated in Figure 3g,h. As NO_2_ acts as an oxidizing agent, its adsorption instigates the generation of hole carriers. In bare graphene, predominant hole carriers escalate channel current, whereas in n-doped graphene, reminiscent of an *n*-type semiconductor in the absence of gate bias, induced p-doping via NO_2_ precipitates a decline in current. Notably, the absolute response hierarchy in the gas sensor manifests as DETA > EDA > TETA > bare, with the DETA-doped sensor demonstrating superior recovery kinetics. EDA, characterized by its elevated vapor pressure, can be effectively purged, while the robust binding of NO_2_ to TETA-doped graphene presents detachment challenges. Conversely, DETA-doped graphene exhibits diminished susceptibility to such adverse effects, thereby exhibiting heightened response and recovery attributes. Remarkably, the exceptional recovery in DETA-doped graphene is ascribed to DETA molecules adeptly masking defects in CVD-grown graphene [36]. To evaluate the stability of the n-doped graphene gas sensors, the devices were assessed following exposure to ambient air for 14 days. The air stability findings are succinctly summarized in Figure 3i. Notably, graphene gas sensors doped with EDA and DETA, recognized for their relatively high response characteristics, were scrutinized alongside a bare graphene gas sensor for comparison. With increasing exposure time to ambient air, both the bare and EDA-doped graphene gas sensors exhibited diminishing responses, with a pronounced decline evident in the EDA-doped variant. Conversely, the DETA-doped graphene gas sensor retained an impressive 70% of its absolute response even after 14 days. This notable resilience is ascribed to the low vapor pressure of DETA, exceeding two-orders-of-magnitude lower than that of EDA. Furthermore, DETA molecules demonstrate robust binding to defects in graphene, thereby bolstering doping stability.

#### 3.1.2. Black Phosphorus (BP)

Black phosphorus (BP), the most thermodynamically stable allotrope of phosphorus, comprises atomic layers held together by van der Waals interactions. Its bulk form resembles graphite and shares the rare distinction of being mechanically exfoliated into ultrathin nanosheets alongside graphite. Bulk BP behaves as a *p*-type semiconductor, boasting a direct band gap of approximately 0.3 eV [37]. However, the band gap of BP can be finely tuned, ranging from 0.3 eV for bulk to 2.0 eV for phosphorene (monolayer BP), contingent upon layer count, suggesting versatile property modulation in the 2D regime [38]. Few-layer phosphorene has demonstrated remarkable performance as a transistor material, showcasing impressive drain current modulation of up to 105 and mobility reaching 1000 cm^2^ V^−1^ s^−1^ [39]. Moreover, it exhibits well-defined current saturation within its transfer characteristics. Leveraging its thickness-dependent direct band gap and expansive surface-to-volume ratio, BP holds promise for applications in gas sensing.

BP nanosheets can be synthesized via mechanical exfoliation, chemical vapor deposition (CVD), and liquid-phase exfoliation methods [40,41,42]. While mechanical exfoliation is straightforward, resulting nanosheets often suffer from oxidation and are limited in size [40]. CVD facilitates the direct synthesis of thin films without the need for bulk preparation. However, this process requires the use of high-purity phosphine gas at elevated temperatures typically ranging from 600 to 1000 °C. The high temperatures and toxicity of phosphine necessitate meticulous handling and management to ensure safety. Furthermore, the thickness and properties of the resulting thin film can vary based on specific growth conditions. Additionally, the CVD process can introduce residual stress due to differences in the thermal expansion coefficients between the thin film and the substrate. This stress can potentially compromise the physical properties of the material, affecting its performance in various applications. Careful optimization of the CVD parameters and post-deposition treatments are essential to mitigate these issues and achieve high-quality thin films with consistent properties [41]. Scaling up BP production for gas sensing presents challenges in cost and safety. Chen and Ho et al. recently demonstrated a high-performance BP nanosheet gas sensor using liquid-phase exfoliation (LPE), exploiting the weak van der Waals forces between BP layers [37,43]. Sonication techniques like probe or bath sonication are commonly used, but a single method may yield irregular nanosheets [44]. To address these limitations, a facile LPE method which is a combination of probe and ice-water bath sonication (as depicted in Figure 4a), has been developed to produce BP nanosheets with precise size control [45]. Initially, bulk BP crystals are fragmented into large sheets with varied shapes and sizes through probe sonication. Subsequently, ice-water bath sonication further shears these large sheets and any remaining unbroken crystals into nanosheets, refining their size distribution. Varying centrifugation speeds were employed to obtain LPE-BP nanosheets, and their morphology was analyzed using TEM imaging. Figure 4b–d illustrate TEM images of multilayer BP-4000, BP-7000, and BP-12000 nanosheets, demonstrating lateral sizes of approximately 1 μm, 600 nm, and 300 nm with thicknesses of 35 nm, 21 nm, and 12 nm, respectively. Notably, increasing centrifugation speed correlates with a reduction in the size of LPE-BP nanosheets.

Figure 4e illustrates the dynamic sensing response of BP-4000, BP-7000, and BP-12000 to NO_2_ gas concentrations ranging from 100 to 1000 ppb at room temperature. The BP gas sensors exhibit response ranges of 8–12% (BP-4000), 141–350% (BP-7000), and 88–388% (BP-12000), respectively. Notably, BP-4000 shows the lowest response across the entire range, attributed to its larger nanosheet size, translating to the reduced specific area and fewer exposed active sites for NO_2_ interaction. At lower NO_2_ concentrations, both BP-7000 and BP-12000 possess sufficient active sites for gas adsorption, with BP-7000 exhibiting a higher response due to its smaller band gap and increased majority carrier concentration, enhancing NO_2_ absorption. Conversely, BP-12000, with a lower carrier concentration, shows a reduced response at lower concentrations. However, at moderate-to-high NO_2_ concentrations, the abundance of active sites becomes paramount, favoring BP-12000 due to its smaller size and higher active site density, leading to superior sensing performance. In Figure 4f, BP-12000 sensors are evaluated at room temperature using various gases, including 100 ppb NO_2_, 1000 ppm NH_3_, 1000 ppm CO, 1000 ppm C_2_H_5_OH, 100 ppm H_2_, and 10 ppm HCHO. The device exhibits significantly higher sensitivity to 100 ppb NO_2_ compared to other gases, attributed to the differences in adsorption energies which are 0.212 eV for NH_3_, 0.055 eV for H_2_, 0.24 eV for C_2_H_5_OH, 0.23 eV for CO, and 0.63 eV for NO_2_. As depicted in Figure 4g, when BP nanosheets encounter NO_2_ gas molecules, the pronounced affinity of NO_2_ prompts the transfer of minority carriers (electrons) from the BP nanosheet to the NO_2_ molecules. This process increases the concentration of main carriers which are holes, thereby augmenting electrical conductivity [46]. Subsequently, NO_2_ molecules, having absorbed electrons from the sensing material, ionize into NO_2_^−^ ions and weakly bond with the BP nanosheets. Recovery ensues through a desorption mechanism, where exposure to N_2_ gas prompts the return of electrons to the nanosheets, restoring the initial resistance value [47]. The remarkable sensing efficacy of the proposed BP nanosheets can be attributed to two primary factors: (1) efficient carrier transport facilitated by the material’s low band gap, and (2) the abundance of active sites made available by the small dimensions of the nanosheets [48]. Given the structural similarity between NO and NO_2_, there is a possibility of sensor response to NO gas as well [49]. This article underscores the potential of phosphorus analogs in gas sensor applications at room temperature, aiming to address the limitations associated with metal-oxide-based chemoresistive gas sensors.

### 3.2. Transition Metal Chalcogenides (TMCs)

#### 3.2.1. WSe_2_

Traditional metal-oxide-based gas sensors typically require high operating temperatures (>150 °C) to facilitate the adsorption and desorption of target gases, resulting in increased production costs and power consumption [50,51,52]. Recently, tungsten diselenide (WSe_2_), a member of the transition metal chalcogenides (TMCs), has garnered significant attention due to its large specific surface and high responsivity to NO_2_ at room temperature. However, most WSe_2_-based gas sensors have predominantly utilized a planar 2D structure, neglecting the higher catalytic activities at the material’s edge. Consequently, these sensors have exhibited low sensitivity and slow reaction kinetics. Enhancing the sensing performance hinges on controlling the material morphology to increase the specific surface area and expose more active edges. Recent studies suggest that the sensitivity of TMC-based sensors can be significantly improved by increasing the edge density through various techniques such as morphology-control defect engineering element doping and heterojunction construction [53,54,55,56,57,58,59].

To enhance catalytic activities, Duan and Zhang et al. reported the development of vertical few-layer 3D-WSe_2_ nanosheets, aiming to increase the number of functional edges and thereby improve sensitivity [60]. The schematic diagram of the 3D-WSe_2_ preparation process is illustrated in Figure 5a. In this process, a TiO_2_ buffer layer is introduced onto a Si/SiO_2_ substrate to facilitate the vertical growth of WSe_2_ layers, thereby exposing more edge sites. The high temperature in the CVD furnace causes the TiO_2_ buffer layer to transform into TiO_2_ nanoparticles. During the growth of WSe_2_, the lattice mismatch with TiO_2_ induces a deviation in the growth direction, leading to the vertical alignment of WSe_2_ nanosheets. This results in the formation of a 3D structure with enhanced exposure of reactive edges. To analyze the morphology of 3D-WSe_2_ nanosheets, SEM imaging is employed. The top and cross-sectional views are presented in Figure 5b,c. The synthesized 3D-WSe_2_ grows vertically, with nanosheets that are connected and uniformly distributed across the substrate surface. These nanosheets have diameters ranging from approximately 1 to 4 μm, with the 3D structure providing more active edge sites, thereby enhancing the adsorption of target gases. The cross-sectional view in Figure 5c reveals that the height of the 3D-WSe_2_ nanosheets varies, remaining below 4 μm due to differences in vertical growth angles. Figure 5d illustrates the working mechanism of the 3D-WSe_2_ sensor. The gas-sensing mechanism based on 2D TMC materials relies on a charge transfer process. When the 3D-WSe_2_ sensor is exposed to NO_2_, gas molecules attach to the WSe_2_ surface, extracting electrons from its valence band and increasing the hole density. For *p*-type WSe_2_, this rise in hole concentration enhances conductivity, leading to a decrease in sensor resistance [61]. Figure 5e illustrates the dynamic sensing response of the 3D-WSe_2_ gas sensor to NO_2_ concentrations ranging from 0.1 to 10 ppm at room temperature [62]. The sensor achieves responses of 5.5, 11.1, 14, 21.6, 34.6, and 47.3% at concentrations of 0.1, 0.2, 0.3, 0.5, 1, and 10 ppm, respectively. Figure 5f presents the response–recovery time curve of the 3D-WSe_2_ gas sensor to 1 ppm NO_2_ at room temperature, showing a response time (τ_res_) of approximately 66 s and a recovery time (τ_rec_) of about 17 min. Gas selectivity and durability are critical for sensor performance. Figure 5g demonstrates the stability of the 3D-WSe_2_ sensor under 1 ppm NO_2_, maintaining a response of 30.5% after 8 weeks, indicating excellent stability. Figure 5h highlights the gas selectivity of the 3D-WSe_2_ sensor exposed to 1 ppm NO_2_, 200 ppm H_2_, 200 ppm NH_3_, and 200 ppm CO at room temperature, with response rates of 34.6%, 3.9%, 9.7%, and 4.1%, respectively. The significantly higher response to NO_2_ is attributed to its high adsorption energy and efficient charge transferability [63,64]. Humidity testing is an essential performance indicator for gas sensors. Figure 5i shows the sensor’s response to 1 ppm NO_2_ at various humidity levels (0%, 20%, 40%, 60%, 80%, and 90%). The responses are 34.6, 31.5, 30.8, 30.7, 27.5, and 27.5%, respectively. The reduction in response with increasing humidity is due to the solubility of NO_2_ in water, reducing its concentration, and the adsorption of water molecules on the active sites of the 3D-WSe_2_, hindering NO_2_ absorption and reaction. However, even at 90% humidity, the sensor maintains 79.4% of its initial response and over 88% at normal humidity levels (11–56%). This article paves a way to improve the gas-sensing performance of 2D TMC-based gas sensor channels at room temperature via nanostructuring for replacing the metal-oxide-based chemoresistive gas sensors.

#### 3.2.2. SnS_2_

Metal-oxide semiconductor gas sensors, such as those based on WO_3_, TiO_2_, MoO_3_, and SnO_2_, are well regarded for their excellent response, stability, and selectivity [65,66,67]. However, these sensors typically require high operating temperatures (>300 °C) to achieve optimal sensing kinetics [68,69]. A promising alternative lies in 2D TMDs such as MoS_2_, WS_2_, and SnS_2_, which feature an adjustable band gap, and high surface activity [70,71]. Among them, SnS_2_ has garnered significant attention for gas-sensing applications due to its abundance on Earth and its suitable intermediate bandgap of 2.35 eV [72,73]. The unpaired outer 5p electrons of SnS_2_ facilitate favorable electron donation/acceptance properties, enabling efficient interactions with a wide variety of gases [74,75,76,77]. Despite its potential, SnS_2_-based gas sensors face challenges such as low sensitivity, slow sensing kinetics, and weak gas selectivity. Recent research efforts have focused on enhancing the sensing properties of SnS_2_ through methods like morphology control, doping, and interface construction [78]. One notable example is the formation of SnS_2_/SnO_2_ heterostructures, where the SnS_2_–SnO_2_ interface has been shown to improve NH_3_ gas detection performance [79,80]. However, the uneven distribution of the interface can hinder the regulation of sensing behaviors and complicate the understanding of interface effects. To address these challenges, researchers have explored doping and nanoarchitecture engineering to adjust the electronic structure and exposed surface of SnS_2_, thereby enhancing gas physisorption and detection characteristics [81]. For instance, Ou et al. synthesized SnS_2_ planar nanodisks with high NO_2_ detection capabilities, attributed to facile physisorption and abundant edge sites [82]. While increasing edge sites can improve gas sensitivity, the largely inactive basal plane in SnS_2_ nanostructures remains underutilized. Given these considerations, advancing the sensing performance of 2D SnS_2_ requires a synergistic approach that integrates both the physical/chemical adsorption mechanisms and defect engineering.

Recently, Bo and Zheng fei et al. reported the synthesis of defective SnS_2_ nanosheets via rapid argon plasma irradiation and investigated their NH_3_ gas-sensing properties [83]. The formation process of these defective SnS_2_ nanosheets is schematically illustrated in Figure 6a. Initially synthesized using a solvothermal method, the SnS_2_ nanosheets are then subjected to a brief argon plasma treatment, creating sulfur vacancy defects on the surface. These vacancies not only provide additional active sites for gas absorption but also adjust the electronic structure of SnS_2_. Figure 6b shows the SEM image of pristine SnS_2_ nanosheets, revealing the ultrathin, smooth nanosheets on the substrate. After 4 s of plasma treatment, the SnS_2_ retains its nanosheet morphology, with some surface roughening and edge passivating due to high-power Ar plasma bombardment, which typically results in anisotropic etching.

The dynamic sensing transients of these four Ar-treated SnS_2_ sensors to 0.2 to 20 ppm of NH_3_ at 125 °C are presented in Figure 6d. The 4 s plasma-treated sample demonstrates superior sensitivity at 125 °C compared to the other sensors, attributed to the increased number of active sites on the basal planes created via plasma treatment, enhancing gas adsorption and interaction. The bar plots in Figure 6e compare the sensing response of these four sensors to 20 ppm of NH_3_ at different operating temperatures. The 3 s plasma-treated SnS_2_ shows a maximum response value of 11.2 at 100 °C.

As the operating temperature increases, the gas response decreases, indicating that desorption of NH_3_ from the surface becomes predominant at higher temperatures. Conversely, the 4 s plasma-treated sample exhibits the highest response at 125 °C and 150 °C compared to other sensors, with Ar plasma-treated SnS_2_ overall showing higher responses than untreated SnS_2_. This enhancement is due to the increased active sites on the basal planes provided by the defective structure from plasma treatment [84]. Compared to untreated SnS_2_, the sensor response of the 4 s plasma-treated sample is increased by 89% for 20 ppm of NH_3_. This enhancement is attributed to the synergy of increased surface area and sulfur vacancies, with vacancies playing a major role [85,86]. Figure 6f demonstrates the selectivity for 20 ppm of CH_4_, CO, acetone, and H_2_ at 125 °C. Comparative results show that the 4 s plasma-treated SnS_2_ sensor exhibits promising NH_3_ selectivity compared to other gases including acetone, H_2_, CO, and CH_4_ [87,88]. This article underscores the potential of designing TMC-based materials with tunable sensing behaviors via defect engineering.

#### 3.2.3. MoS_2_

MoS_2_, a standout material in the TMD family, exhibits remarkable performance in various applications such as FETs [89], p–n junctions [90], and photodetectors [91], thanks to its direct band gap of 1.8 eV in a single layer. Recent studies have demonstrated the efficient performance of MoS_2_-based gas sensors at ppm level [92], and, notably, these sensors are capable of detecting NO_2_ gas at sub-ppm level [93]. However, these sensors typically suffer from slow response and poor recovery at room temperature. Subsequent reports have shown improved recovery by utilizing elevated temperatures ranging from 100 to 150 °C, which accelerate the desorption of gas molecules [94]. Another alternative approach involves illuminating the MoS_2_ channel, which enhances both sensitivity and recovery [95].

Pham and Mulchandani et al. reported significantly enhanced gas sensitivity in MoS_2_ gas sensors under red light illumination [96]. Figure 7a presents an optical image of the MoS_2_ optoelectronic gas sensor with Au electrodes with a 10 μm gap. Figure 7b,c show SEM images of the MoS_2_ layer forming the sensing channel, exhibiting the typical triangular morphology of single-layer MoS_2_ crystals. In Figure 7d, the NO_2_ detection capability of the MoS_2_-based gas sensor with Au electrodes in the dark and under red LED illumination (incident power of 60.9 nW and light intensity of 60.9 mW/cm^2^) is displayed. Under red LED illumination, a NO_2_ sensitivity of S = 4.9%/ppb (4900%/ppm) was observed, approximately 50-times higher than the best value of 0.1%/ppb obtained in the dark. Moreover, devices with graphene and graphene/Au electrodes were evaluated for NO_2_ gas sensing under light illumination. Figure 7e shows the transient response curve of the MoS_2_ channel with graphene electrodes (Gr-MoS_2_-Gr) to NO_2_ gas exposure at concentrations ranging from 25 to 200 ppb. The device exhibited a response of ~ 16% for 25 ppb NO_2_. For devices with a MoS_2_ channel with a graphene electrode protected by an Au layer (Figure 7f), a similar trend with a much higher magnitude and faster, complete recovery upon N_2_ exposure was observed. Figure 7g shows the band diagram of the MoS_2_ channel in the dark. For *n*-type doped MoS_2_, the Fermi level shifts toward the conduction band. Due to the relatively large band gap of MoS_2_ and high Schottky barriers at the metal electrode/MoS_2_ interfaces, a low concentration of free charge carriers could be formed [97]. Thermally generated electrons and holes can act as the majority and minority carriers. The relatively low sensitivity of the sensor in the dark can be explained by the presence of oxygen, which traps thermally generated electrons in the MoS_2_ channel, reducing the device current and blocking most electrons from interacting with NO_2_ molecules, leading to a suppressed response to the target gas. As schematically illustrated in Figure 7h, under red light illumination, the concentration of free electrons increases by several orders of magnitude [95]. This study highlights a breakthrough in MoS_2_ gas sensor applications by using light illumination to address the issue of high operating temperatures in gas sensors based on TMCs.

### 3.3. Hybrid Structure

#### 3.3.1. SnS_2_/In_2_S_3_ Heterostructure

Recent research has extensively investigated the gas-sensing properties of SnS_2_-based sensors at different operating temperatures. For instance, well-dispersed 2D SnS_2_ flakes obtained via wet chemical routes exhibited high selectivity and reversibility for NO_2_ gas in the range of 0.6–10 ppm at 120 °C, attributed to their planar structure and physisorption-based charge transfer. Similarly, ultrathin SnO_2_/SnS_2_ heterojunction nanoflowers, prepared via in situ oxidation, displayed a strong response to NO_2_ at 60 °C, relying on the large number of localized electrons at the SnO_2_–SnS_2_ interface [83,98]. Despite these advancements, there is a need for efficient sensors with high surface molecular recognition capacity and superior gas selectivity to other harmful gases, especially at low temperatures. For example, triethylamine (TEA) is a commonly used organic solvent known for its pungent odor, which poses health risks upon volatilization [99]. Efforts have been made to develop TEA-sensing sensors, such as SnS_2_/ZnS microspheres, which showed fast response/recovery times to 50 ppm TEA at 180 °C [100]. However, operating at high temperatures can increase power consumption and production costs, as well as pose physicochemical instability issues. Therefore, achieving near-room-temperature TEA-sensing detection with high response and gas selectivity requires optimization of synthetic technology, effective heterojunctions, and modification of surface/interface transmission mechanisms. 

In_2_S_3_, another 2D layered semiconductor, has shown sensitivity to TEA and can be combined with SnS_2_ easily to form heterostructures, potentially enhancing gas-sensing capabilities. Wang and Ma et al. reported flower-like self-assembled SnS_2_/In_2_S_3_ composites with higher sensitivity and selectivity, lower operating temperatures, and faster response/recovery times compared to pristine SnS_2_ [101]. These composites are prepared via a simple hydrothermal method with varying molar ratios (0.5%, 1%, and 3%) of InCl_3_ added to the precursor solution as illustrated in Figure 8a. This process resulted in composites labeled for SnS_2_–0.5% In, SnS_2_–1% In, and SnS_2_–3% In. The responses of these samples to different concentrations of TEA are shown in Figure 8b. The optimal operating temperatures of SnS_2_ and SnS_2_–0.5% In are determined to be 50 °C and that of the SnS_2_–1% In and SnS_2_–3% In operated optimally at 40 °C, determined via the adsorption and excitation capacity of TEA and oxygen molecules due to the concentration of heterostructures of SnS_2_/In_2_S_3_. The responses of all sensors increased consistently with increasing TEA concentrations from 10 to 200 ppm. The SnS_2_/In_2_S_3_ heterostructures exhibited up to five-times higher responses (SnS_2_–1% In) than those of pristine SnS_2_. For instance, the gas responses to 200 ppm TEA for the SnS_2_ and SnS_2_–1% In sample were 5.89 and 26.45 at operating temperatures of 50 °C and 40 °C, respectively. Figure 8c shows that relationship curves of the response versus TEA concentration exhibit good linearity, with the SnS_2_–1% In sensor displaying a superior linear relationship with a slope of 0.6182, indicating its potential for accurate gas concentration detection. Figure 7e and Figure 8d illustrate the selectivity of various samples to 100 ppm of different gases. Both pristine SnS_2_ and SnS_2_/In_2_S_3_ heterostructure samples show a high selectivity to the TEA gas compared to the other gases such as alcohols, acetone, ammonia, etc. Especially, the SnS_2_–1% In sample exhibits a twenty-fold difference in responses between TEA and methanol, showcasing its enhanced sensitivity. The improved gas sensitivity and selectivity of SnS_2_/In_2_S_3_ heterostructures, compared to pristine SnS_2_, primarily result from their larger surface area, increased reactive sites, and enhanced adsorption capacity due to S vacancies. Nitrogen-based gases like TEA can chemically bond to the molecular layer via strong covalent bonds. Upon exposure to the test gas, TEA readily adsorbs onto the surface of the sensing material and undergoes redox reactions. This process is facilitated by SnS_2_–In_2_S_3_ heterojunctions, which enhance TEA adsorption and electronic transmission efficiency. Furthermore, the bond energies of various gases, such as ethanol (458.8 kJ/mol), acetone (798.9 kJ/mol), TEA (307 kJ/mol), and formaldehyde (411 kJ/mol), play a crucial role in gas selectivity. The low bonding energy of TEA in particular facilitates its involvement in redox reactions with sensitive materials during gas-sensing processes, thereby enhancing the selectivity and sensitivity of the fabricated gas sensor. Nitrogen-based gases like TEA could form strong covalent bonds with the molecular layer, resulting in their chemical adsorption on the sensing material. The formation of SnS_2_–In_2_S_3_ heterojunctions enhances TEA adsorption and electron transmission efficiency, facilitating redox reactions during gas sensing [102]. Furthermore, the formation of chemical bonding energies of gases plays a crucial role in gas selectivity, with TEA’s lower bond energy favoring its oxidation–reduction reaction with sensing materials. The practical application of SnS_2_/In_2_S_3_ composites for detecting and quantifying low concentrations of TEA gas released during shrimp storage is depicted in Figure 8f. As seafood ages, the amount of gaseous organic amines such as TEA increases gradually. Gas samples of 5 mL were periodically extracted from shrimp-stored containers for gas-sensing analysis. The responses of SnS_2_–1% In to gas emitted at different storage durations (12, 24, 36, 48, and 60 h) are recorded as 1.32, 2.02, 3.06, 4.05, and 7.14, respectively. The flower-like SnS_2_/In_2_S_3_ composite forms a 2D layered structure with adjustable folds on its surface, providing extensive and abundant active sites crucial for enhancing both gas adsorption and electron transmission processes. The creation of SnS_2_–In_2_S_3_ heterojunctions directly facilitates electron transport across the interface. Upon the combination of SnS_2_ and In_2_S_3_, an internal electric field is generated at these heterogeneous interfaces. Thermally activated electrons preferentially transfer from the conduction band of SnS_2_ to In_2_S_3_ due to this built-in electric field, enabling their involvement in surface redox reactions. During electron transfer, the energy bands of SnS_2_ and In_2_S_3_ undergo bending until reaching a new equilibrium Fermi level. This process allows the formation of a new hole accumulation layer. Gas molecules interact rapidly with O^δ−^ species on the surface, leading to the release of trapped electrons back into the heterostructure. This mechanism underscores the role of SnS_2_/In_2_S_3_ heterojunctions in promoting efficient gas sensing through enhanced electron transfer dynamics and surface reactivity. This study provides the potential of designing an SnS_2_-based heterostructure gas sensor that operates at low temperatures, fabricated via a simple hydrothermal method.

#### 3.3.2. MoS_2_/ZnO Heterostructure 

MoS_2_ offers promising advantages for developing RT sensors [103], attributed to its active sulfur sites, facile functionalization, and compatibility with large-scale synthesis techniques. Although the single- and few-layer MoS_2_ sensors have been demonstrated to exhibit selective NO_2_ detection with a low detection limit (LoD) [104], they showed low gas response, slow kinetics, and poor recovery at RT which are attributed to the prevalent use of horizontally aligned MoS_2_ with limited dangling bonds, roughness, and exposed edge sites [105]. New approaches to address these limitations have been explored, including ligand conjugation, chemical doping, and surface functionalization with noble metals and other nanomaterials [106,107].

Gond and Rawat et al. recently reported a facile method to fabricate the heterostructures using vertically aligned MoS_2_ (VA-MoS_2_) and ZnO for NH_3_ gas sensors via the CVD method. The VA-MoS_2_ flakes were synthesized via a three-heating zone CVD method. Initially, an Ar flow of 300 sccm was maintained to remove moisture from the precursor surface. Subsequently, the temperatures required for precursor evaporation and MoS_2_ synthesis were achieved: 680 °C for MoO_3_ powder and 200 °C for sulfur powder. During this phase, the Ar flow was reduced to 100 sccm. Finally, to enhance surface reactions and facilitate the synthesis of VA-MoS_2_, the Ar flow was further decreased to 25 sccm. This controlled reduction in Ar flow played a critical role in achieving the desired structure and properties of the VA-MoS_2_ flakes [108]. They showed the high potential of VA-MoS_2_/ZnO composites by exploring gas sensor performances such as gas response, selectivity, and response/recovery time. Figure 9a illustrates the structural schematic of the VA-MoS_2_/ZnO gas sensor channels with Ag electrodes via the e-beam evaporation method. Figure 9b,c present FESEM images of VA-MoS_2_ and VA-MoS_2_/ZnO heterostructures, showcasing their surface morphology. CVD-grown VA-MoS_2_ displays a densely interconnected network of flakes on a SiO_2_/Si substrate, with numerous exposed edge sites. The FESEM image of the VA-MoS_2_/ZnO heterostructure reveals a uniform distribution of spherical ZnO nanoparticles (NPs), approximately 20 nm in size, over the MoS_2_ flakes. Figure 9d illustrates the dynamic sensing response of the VA-MoS_2_ gas sensor to NH_3_ gas concentrations ranging from 5 to 50 ppm at room temperature. The baseline resistance of the fabricated gas sensor in dry air is approximately 25.84 KΩ. As NH_3_ concentration increases, the resistance of the device rises significantly, showing a notable response of 9.3% even at a low concentration of 5 ppm. This substantial response suggests the potential for further optimization to develop a highly sensitive sensor for detecting low-level NH_3_ concentrations. In Figure 9e, both the response and recovery times of the sensor increase remarkably with rising NH_3_ gas concentration. Specifically, within the 5 to 50 ppm range, the response time increases from 138 s to 570 s, and its recovery time increases from 155 s to 972 s. This behavior can be attributed to the active sites on the sensor material saturating more rapidly at higher gas concentrations, leading to slower absorption and desorption processes. Figure 9f demonstrates the response of the VA-MoS_2_/ZnO sensor to 50 ppm NH_3_ at room temperature (RT) as relative humidity (RH) increases from 5% to 75%. The sensor maintains high responses of 88.86%, 87.9%, and 75.25% at RH levels of 25%, 45%, and 75%, respectively, indicating effective NH_3_ detection even at high RH levels. The decrease in response with increasing RH is due to hindered charge carrier transportation and partial obstruction of reaction sites by water molecules. To quantify the gas selectivity, Figure 9g shows the response of the VA-MoS_2_/ZnO heterostructure for 50 ppm of NH_3_, NO_2_, SO_2_, and CO and 1000 ppm of CO_2_ gases. Although the VA-MoS_2_/ZnO channel can react with various gases, the most significant gas response of 100.4% is observed for NH_3_, with the second-highest response being approximately 45.18% for NO_2_. This elevated response to NH_3_ compared to the other target gases highlights the enhanced selectivity of the VA-MoS_2_/ZnO sensor towards NH_3_ at RT. This exceptional selectivity is attributed to differences in chemical reactivities among the target gases, stemming from variations in bonding energies and chemical molecular structures. This article presents a breakthrough in TMD material-based gas sensor applications, addressing the challenge of low recoverability at RT via the utilization of vertically aligned structures and heterostructure with other metal-oxide semiconductors.

## 4. Perspective and Challenges

2D materials have emerged as advantageous for gas detection due to their high specific surface area and active surface dangling bonds. Compared to conventional metal-oxide gas sensors, 2D gas sensors offer a higher surface-to-volume ratio, facile surface functionalization, and layered structures that facilitate the control of surface chemical reactions through heterojunctions. Despite these advantages, 2D gas sensors have faced challenges such as lower gas response and slower response/recovery rates than metal-oxide-based gas sensors. Addressing these issues has been the focus of extensive research, with numerous studies demonstrating the potential of 2D-material-based gas sensors. However, the problems of low gas sensitivity and slow response/recovery times still require further improvement. In this review, we have summarized three primary approaches to address these challenges: (1) the formation of nanostructures, (2) the development of heterostructures, and (3) the application of photoactivation techniques. These strategies represent promising avenues for enhancing the performance of 2D gas sensors and realizing their full potential in practical applications.

First, the nanostructures enhance gas sensor performance by addressing the slow response and desorption rates through several key mechanisms. They provide a high surface-area-to-volume ratio, increasing the number of active sites for gas interactions, and facilitating rapid gas diffusion due to their small size and porosity. The increased surface energy of the nanostructure strengthens the gas adsorption and desorption, while optimized electron transport pathways improve response times. Additionally, the presence of crystal defects in nanostructures facilitates extra active sites for gas adsorption/desorption, further enhancing gas molecule interactions. These factors collectively lead to faster response and recovery times, improved sensitivity, and better gas-sensing capabilities.

Second, the heterostructures combine different materials to increase catalytic activity and adsorption sites, promoting faster gas molecule interactions. The formation of charge carrier depletion layers at interfaces and the Fermi level-mediated charge transfer creates potential barriers that amplify sensor response. Synergistic effects between materials and tunable junction properties allow for optimized sensitivity and selectivity. These improvements are attributed to the enhanced reaction kinetics, leading to faster response and recovery times, and enhanced sensitivity across various environmental conditions.

Finally, photoactivation promotes oxygen ion interactions, reduces activation energy, enables room temperature operation, and generates electron–hole pairs. The generated exertion pairs could improve the gas molecule adsorption, selectivity, and response/recovery times. Ultimately, photoactivation allows efficient gas detection specifically under room temperature for 2D materials, which have optical band gaps with a visible range, without requiring high operating temperatures.

In summary, we have explored the advancements and challenges in gas sensors based on two-dimensional materials. We believe this comprehensive overview will provide valuable insights into the current state of the 2D-material-based chemoresistive gas sensors, highlighting innovative strategies for performance enhancement. Furthermore, we hope this review could be a useful resource for researchers and engineers working towards the next-generation gas-sensing devices, inspiring future innovations in this rapidly evolving area. (Table 1).

## Figures and Tables

**Figure 1 nanomaterials-14-01397-f001:**
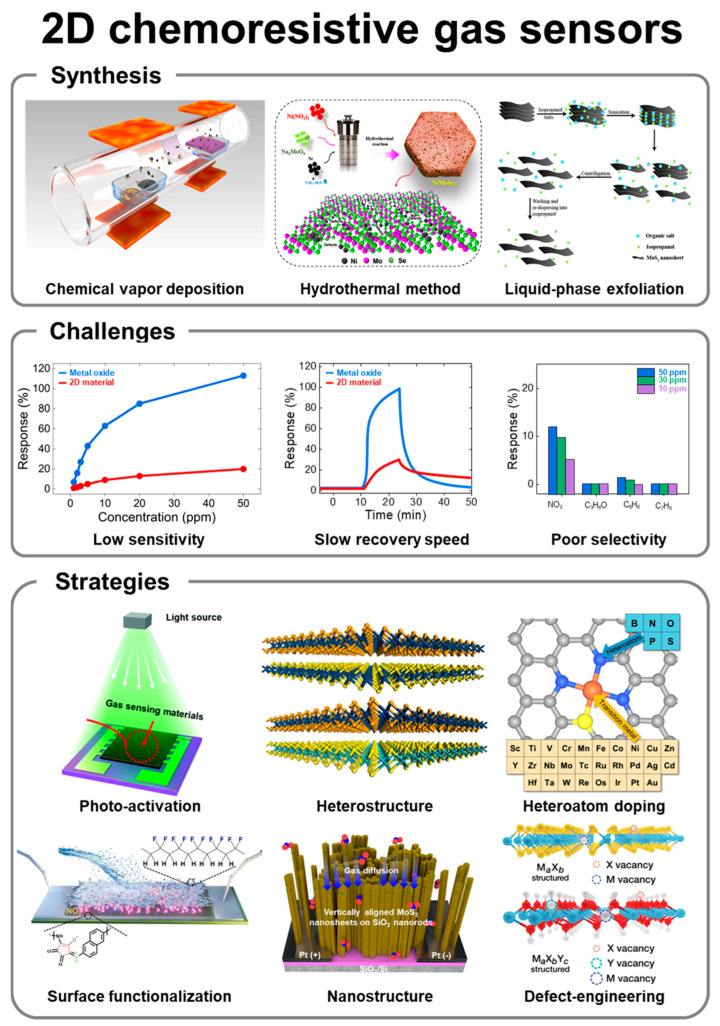
Overview of 2D chemoresistive gas sensors. Reprinted from [9,10,11,12,13,14,15,16,17].

**Figure 2 nanomaterials-14-01397-f002:**
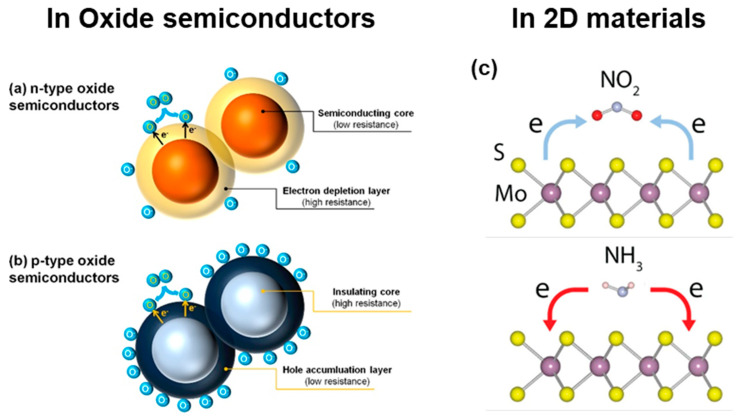
Gas-sensing mechanisms in (**a**) *n*-type semiconductor, (**b**) *p*-type semiconductor, and (**c**) 2D-materials-based gas sensors. Reprinted from [19,20].

**Figure 3 nanomaterials-14-01397-f003:**
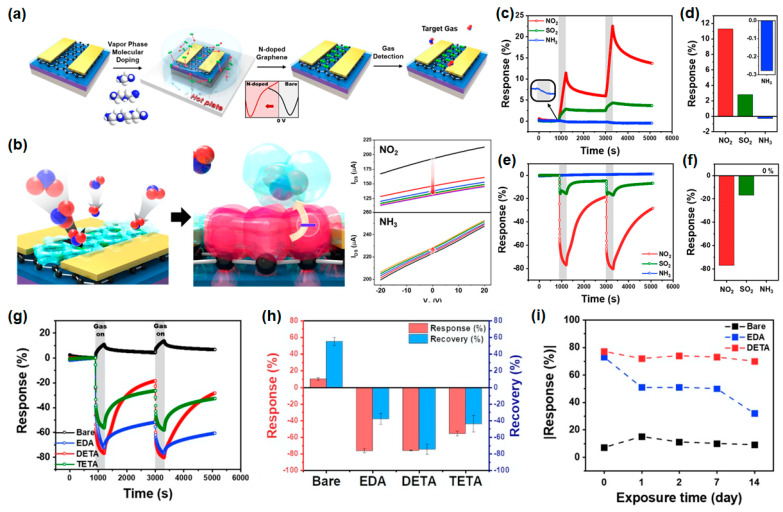
(**a**) Schematic illustration of the fabrication of a gas sensor based on *n*-doped graphene FETs with EDA, DETA, and TETA. (**b**) Sensing mechanism of n-doped graphene to NO_2_. (**c**) Gas selectivity and (**d**) summarized response of pristine graphene. (**e**) Gas selectivity and (**f**) summarized response of DETA-doped graphene. (**g**–**i**) Gas sensor performance of n-doped graphene: (**g**) NO_2_ sensing properties of bare, EDA-, DETA-, and TETA-doped graphene, (**h**) the extracted response and recovery, and (**i**) stability characteristics by exposing the devices to atmosphere. Reprinted from [33].

**Figure 4 nanomaterials-14-01397-f004:**
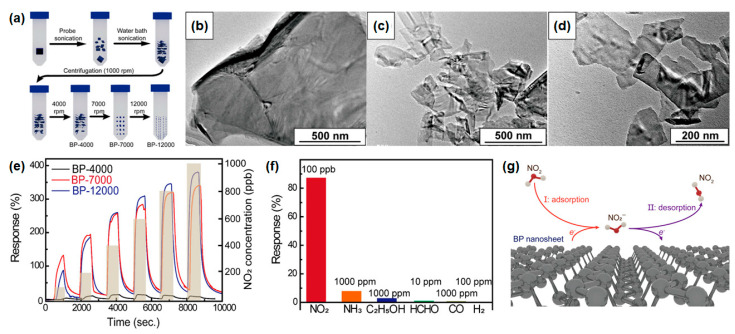
(**a**) Schematic illustration of BP nanosheet synthesis process through liquid-phase exfoliation via combined probe and bath sonication steps. TEM image of (**b**) BP-4000, (**c**) BP-7000, and (**d**) BP-12000. (**e**) Dynamic gas response of BP nanosheets under NO_2_ gas. (**f**) Gas-selectivity test of BP-12000 under various gases. (**g**) Schematic of the NO_2_ sensing mechanism of BP nanosheets. Reprinted from [45].

**Figure 5 nanomaterials-14-01397-f005:**
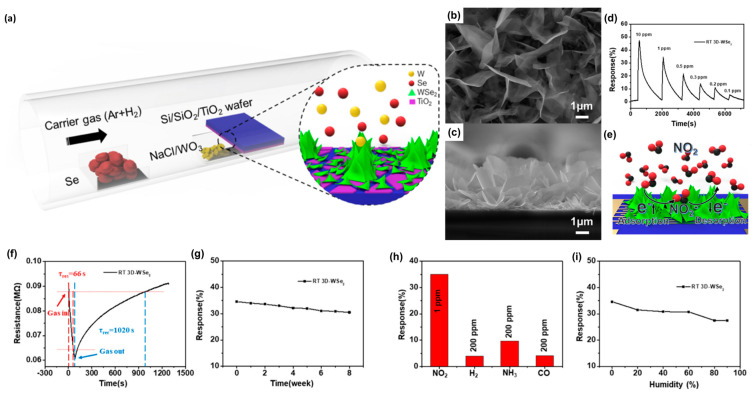
(**a**) Schematic illustration of 3D-WSe_2_ synthesis process through the chemical vapor deposition method. (**b**) SEM top-view image and (**c**) cross-sectional view of 3D-WSe_2_. (**d**) Schematic diagram of the mechanism of the 3D-WSe_2_ gas sensor. (**e**–**i**) Gas-sensor properties of 3D-WSe_2_: (**e**) NO_2_-sensing property, (**f**) response–recovery time curve, and stability characteristics by exposing devices to 1 ppm NO_2_ gas. (**h**) Gas-selectivity test and (**i**) humidity test. Reprinted from [60].

**Figure 6 nanomaterials-14-01397-f006:**
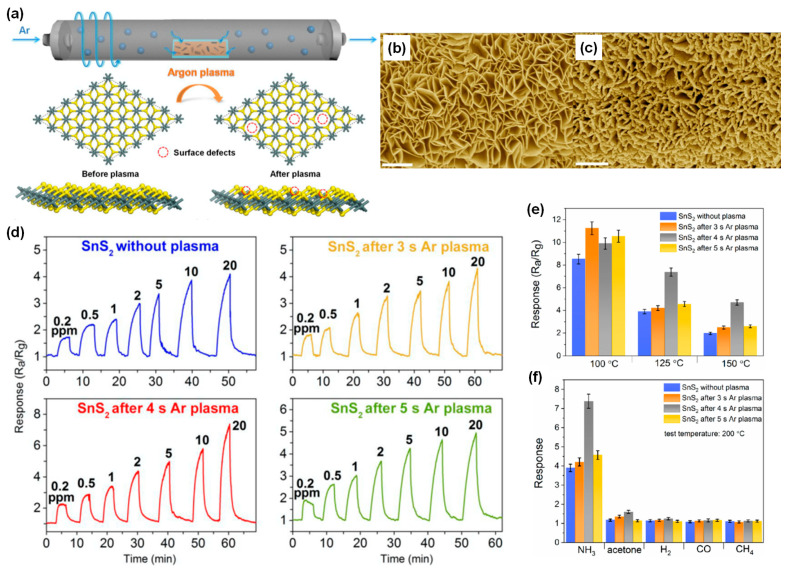
(**a**) Schematic illustration of Ar plasma-treated SnS_2_. SEM image of SnS_2_ (**b**) without plasma treatment and (**c**) after 4 s of Ar plasma treatment. Dynamic gas response under NH_3_ gas of (**d**) SnS_2_ without Ar treatment, and SnS_2_ after 3 s, after 4 s, and after 5 s of Ar plasma treatment. Comparison of gas-sensing properties of different SnS_2_ samples. (**e**) Gas response and (**f**) gas selectivity. All gas concentrations are 20 ppm. Reprinted from [83].

**Figure 7 nanomaterials-14-01397-f007:**
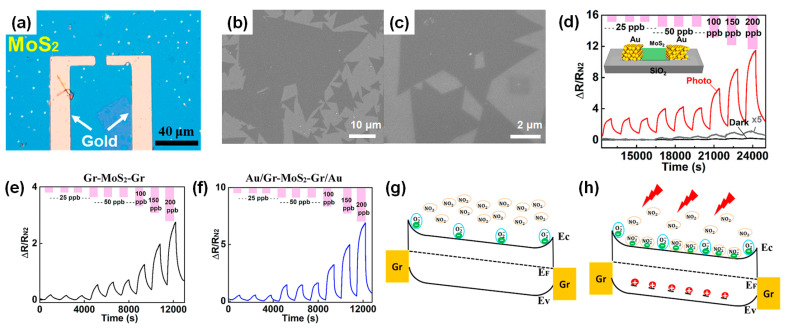
(**a**) Optical image of MoS_2_ device with Au electrodes. (**b**,**c**) SEM images of layered-MoS_2_ at different magnifications. (**d**) Response curve of Au-MoS_2_-Au device toward 25 to 200 ppb NO_2_ gas in the dark (black line; gray line shows 5-fold magnified data) and under red LED illumination (red curve). Dynamic response curve of (**e**) Gr-MoS_2_-Gr and (**f**) Au/Gr-MoS_2_-Gr/Au devices. Band diagram of the device showing interaction of conduction band electrons in MoS_2_ with NO_2_ gas molecules (**g**) in the dark and (**h**) under red light illumination. Reprinted from [96].

**Figure 8 nanomaterials-14-01397-f008:**
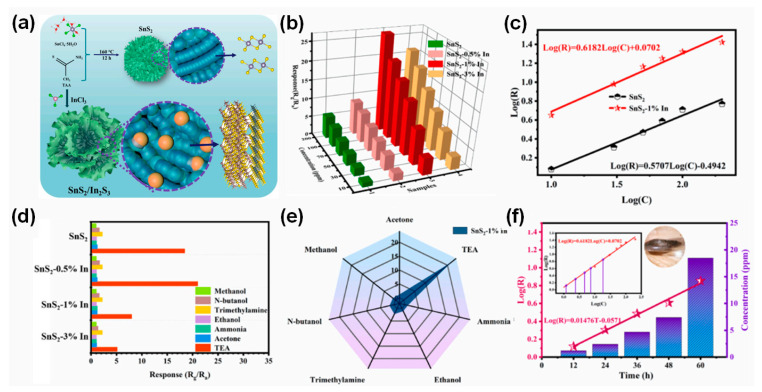
(**a**) Schematic diagram of the preparation process of SnS_2_ and SnS_2_/In_2_S_3_ samples. (**b**) Response of SnS_2_/In_2_S_3_ samples to different gas concentrations of TEA, and (**c**) the Log(R) vs. Log(C) curve of SnS_2_ and SnS_2_–1% In_2_S_3_. (**d**,**e**) Selectivity of SnS_2_/In_2_S_3_ heterostructure gas sensor devices under 100 ppm of various gases. (**f**) Linear relationships of the responses’ different seafood storage times, and corresponding concentrations (the inset represents the relationship curves of Log(R) vs. Log(C) of SnS_2_–1% In_2_S_3_). Reprinted from [101].

**Figure 9 nanomaterials-14-01397-f009:**
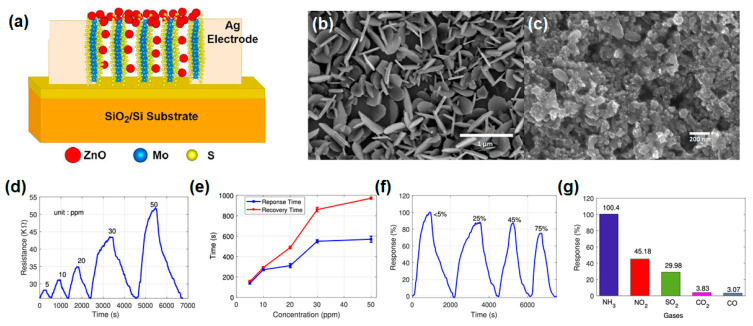
(**a**) Structure schematic of VA-MoS_2_/ZnO device. (**b**,**c**) FESEM image of (**b**) MoS_2_ and (**c**) MoS_2_/ZnO heterostructure. Gas-sensing properties of VA-MoS_2_/ZnO heterostructure: (**d**) dynamic resistance curve, (**e**) response and recovery times, (**f**) humidity test under NH_3_ gas, and (**g**) selectivity under 50 ppm of various gases. Reprinted from [108].

**Table 1 nanomaterials-14-01397-t001:** Comparison table of 2D-material-based chemoresistive gas sensor.

Materials	Temperature (°C)	Target Gas	Concentration (ppm)	Response (%)	Response Time (s)	Recovery Time (s)	Ref.
Graphene	RT	NO_2_	150	30	240	-	[109]
rGO	RT	NO_2_	0.6	2	116	169	[110]
graphene	RT	NO_2_	50	77	-	-	[26]
MoS_2_	RT	NH_3_	2	0.3	15	-	[111]
MoS_2_	RT	NO_2_	0.025	16	-	-	[86]
WS_2_	RT	NO_2_	1	34.6	66	1020	[47]
SnS_2_	100	NH_3_	20	1120	-	-	[62]
SnS_2_/In_2_S_3_	40	TEA	200	2645	-	-	[95]
MoS_2_/ZnO	RT	NH_3_	5	9.3	138	155	[100]

## Data Availability

No new data were created or analyzed in this study.

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
