# Peer review of "Recent Advances in Chemoresistive Gas Sensors Using Two-Dimensional Materials"

_nanomaterials, 2024, doi:10.3390/nano14171397_

Round 1

Reviewer 1 Report

Comments and Suggestions for Authors

This is a timely and good review for the 2D materials based gas sensor. I recommend its publication after addressing the following issue. 

1. This paper focused on the chemoresistive gas sensors with 2D materials. It is suggested to add one figure to explain the chemical sensoring mechanism in the Part 2.1, for better understanding. 

2. For the introduction of hybrid structure of 2D materials for chemical sensor, it is suggested to classify by structure, not only two case of materials. For example, 2D-2D, 2D-1D, 2D-0D. Different construction modes also yields different electrochemical performance, such as 2D-0D including 0D particles on 2D (3.3.2. MoS2/ZnO heterostructure), or 0D core-2D shell (Sci. Rep. 2014, 4, 6745; Adv. Mater. 2021, 33, 2008024).

Comments on the Quality of English Language

Good

Author Response

  1. This paper focused on the chemoresistive gas sensors with 2D materials. It is suggested to add one figure to explain the chemical sensoring mechanism in the Part 2.1, for better understanding.

We appreciate the reviewer for the insightful comment. As the reviewer mentioned, we have inserted one figure to explain the exact sensing mechanism in metal-oxide-based semiconductors and 2D materials-based ones in the Part 2.1. The following figure have inserted in our revised manuscript.

Page 4

Figure 2. Gas sensing mechanisms in (a) n-type semiconductor, (b) p-type semiconductor, and (c) 2D materials-based gas sensors. Reprinted from [19,20]

Page 5, Line 136-138

à The schematic illustration of these working mechanisms of both metal-oxide semicon-ductors and 2D materials-based gas sensors is summarized in Figure 2.

  1. For the introduction of hybrid structure of 2D materials for chemical sensor, it is suggested to classify by structure, not only two case of materials. For example, 2D-2D, 2D-1D, 2D-0D. Different construction modes also yields different electrochemical performance, such as 2D-0D including 0D particles on 2D (3.3.2. MoS2/ZnO heterostructure), or 0D core-2D shell (Sci. Rep. 2014, 4, 6745; Adv. Mater. 2021, 33, 2008024).

We appreciate the reviewer’s comments. However, the focus of this manuscript is to highlight and understand recent advancements in 2D materials-based gas sensor, rather than to deeply investigate the heterojunction of 2D materials. We have already included discussions on notable examples of heterostructures, such as the 2D SnS2-2D In2S3 and 0D ZnO-2D MoS2 structures, to illustrate their roles and advantages in gas sensing. While the suggested papers offer valuable sights into heterostructures involving 2D materials and other components, we believe that our current selection sufficiently demonstrates the enhancement of gas sensing properties achieved through heterostructure formation.

Reviewer 2 Report

Comments and Suggestions for Authors

The authors  summarize the various approaches to enhance the 2D material-based gas sensors and provides an overview of their progress. The review is well written providing critical comments and discussion of the different 2D-materials and their performance with relative merits. The comparisons with metal oxide-based gas sensors are good, and the problems of low sensitivity with 2D-materials are discussed. The review would benefit from an assessment of the progress towards commercially available 2D material gas sensors, as all of these reviewed are still in the research stage. Eg. companies such as ParaGraf and others are now emerging to exploit potential sensor markets although these seem to be still undefined worldwide.

Author Response

Reviewer #2

----------------------------------------------------------------------------------------------------------------

The authors  summarize the various approaches to enhance the 2D material-based gas sensors and provides an overview of their progress. The review is well written providing critical comments and discussion of the different 2D-materials and their performance with relative merits. The comparisons with metal oxide-based gas sensors are good, and the problems of low sensitivity with 2D-materials are discussed. The review would benefit from an assessment of the progress towards commercially available 2D material gas sensors, as all of these reviewed are still in the research stage. Eg. companies such as ParaGraf and others are now emerging to exploit potential sensor markets although these seem to be still undefined worldwide.

----------------------------------------------------------------------------------------------------------------

We sincerely thank the reviewer for the insightful comment. We totally revised the inappropriate and ambiguous sentences to deliver the clear and specific meanings. We also did English language correction for the wider readership. According to this comment, we have revised the manuscript.

Reviewer 3 Report

Comments and Suggestions for Authors

This review deals with advancements in chemiresistive gas sensors using 2D materials, mainly graphene, black phosphorus, transition metal chalcagenides and hybrid structures. A comprehensive survey of sensing mechanisms for n-type and p-type materials has been provided. Strategies to improve 4S (sensitivity, selectivity, speed, stability) approach of gas sensing materials have been depicted as well. The results are very interesting and well presented. Some Minor Revisions are suggested:

- a Table of gas sensing data should be provided to compare performance of 2D materials as reported in literature;

- all figures include a subset of figures at low resolution. This limits the visibility of data reported there. Please, increase resolution of sub-figures to appreciate data and plots;

- page 1. Line 33: "catalytic palliator" should be changed into "pellistor";

- page 9. Line 320-322. Please, check the thickness of BP-layers as reported in manuscript. Please, amend thickness, if any.

After these Minor Revisions, the manuscript should be publishable.

Comments on the Quality of English Language

Minor editing of English should be beneficial.

Author Response

Reviewer #3

----------------------------------------------------------------------------------------------------------------

This review deals with advancements in chemiresistive gas sensors using 2D materials, mainly graphene, black phosphorus, transition metal chalcagenides and hybrid structures. A comprehensive survey of sensing mechanisms for n-type and p-type materials has been provided. Strategies to improve 4S (sensitivity, selectivity, speed, stability) approach of gas sensing materials have been depicted as well. The results are very interesting and well presented. Some Minor Revisions are suggested:

After these Minor Revisions, the manuscript should be publishable.

- a Table of gas sensing data should be provided to compare performance of 2D materials as reported in literature;

- all figures include a subset of figures at low resolution. This limits the visibility of data reported there. Please, increase resolution of sub-figures to appreciate data and plots;

- page 1. Line 33: "catalytic palliator" should be changed into "pellistor";

- page 9. Line 320-322. Please, check the thickness of BP-layers as reported in manuscript. Please, amend thickness, if any.

----------------------------------------------------------------------------------------------------------------

  1. A table of gas sensing data should be provided to compare performance of 2D materials as reported in literature;

We sincerely thank the reviewer for the fruitful comment. As the reviewer mentioned, we provided the performance summary of 2D materials-based gas sensors as a table in Part 4. According to this comment, we have revised the manuscript as below.

Materials

Temperature

(°C)

Target

gas

Concentration

(ppm)

Response

(%)

Response time

(s)

Recovery time

(s)

Ref.

Graphene

RT

NO2

150

30

240

-

[109]

rGO

RT

NO2

0.6

2

116

169

[110]

graphene

RT

NO2

50

77

-

-

[26]

MoS2

RT

NH3

2

0.3

15

-

[111]

MoS2

RT

NO2

0.025

16

-

-

[86]

WS2

RT

NO2

1

34.6

66

1020

[47]

SnS2

100

NH3

20

1120

-

-

[62]

SnS2/In2S3

40

TEA

200

2645

-

-

[95]

MoS2/ZnO

RT

NH3

5

9.3

138

155

[100]

Page 18, Line 689

Table 1. Comparison table of 2D material-based chemoresistive gas sensor.

  1. all figures include a subset of figures at low resolution. This limits the visibility of data reported there. Please, increase resolution of sub-figures to appreciate data and plots;

We are grateful for the reviewer`s valuable feedback. As the reviewer mentioned, we have increased the resolution of all sub-figures to enhance the visibility of data and plots. The updated figures are included in the revised manuscript.

  1. Page 1. Line 33: "catalytic palliator" should be changed into "pellistor";

We appreciate the reviewer`s comment. As the reviewer mentioned, we adjusted it to “pellistor” in the introduction. According to this comment, we have revised the manuscript.

Page 1, Line 33

à Various gas sensors have been developed including optical, electrochemical, pellistor, field-effect transistor (FET)-type, and chemoresistive gas sensors.

  1. Page 9. Line 320-322. Please, check the thickness of BP-layers as reported in manuscript. Please, amend thickness, if any.;

We appreciate reviewer`s careful review. As the reviewer mentioned, we have checked and updated the thickness of the BP-layers. According to this comment, we have revised the manuscript.

Page 9, Line 325-327

à Figures 4b-d illustrate TEM images of multilayer BP-4000, BP-7000, and BP-12000 nanosheets, demonstrating lateral sizes of approximately 1 μm, 600 nm, and 300 nm with thicknesses of 35 nm, 21 nm, and 12 nm, respectively.
